# Tolerability and Effectiveness of Cathodal Transcranial Direct Current Stimulation in Children with Refractory Epilepsy: A Case Series

**DOI:** 10.3390/brainsci13050760

**Published:** 2023-05-04

**Authors:** Soumya Ghosh, Lakshmi Nagarajan

**Affiliations:** 1Children’s Neuroscience Service, Department of Neurology, Perth Children’s Hospital, Nedlands, WA 6009, Australia; lakshmi.nagarajan@health.wa.gov.au; 2Perron Institute for Neurological and Translational Science, University of Western Australia, Nedlands, WA 6009, Australia; 3School of Medicine, University of Western Australia, Nedlands, WA 6009, Australia

**Keywords:** transcranial direct current stimulation, drug-resistant epilepsy, children, seizure frequency, EEG epileptiform abnormality

## Abstract

There are limited treatment options for drug-resistant epilepsy (DRE) in children. We performed a pilot study to investigate the tolerability and effectiveness of cathodal transcranial direct current stimulation (tDCS) in DRE. Twelve children with DRE of varied etiology underwent three to four daily sessions of cathodal tDCS. The seizure frequency at 2 weeks before and after tDCS was obtained from seizure diaries; clinic reviews at 3 and 6 months assessed any longer-term benefits or adverse effects. The spike wave index (SWI) was analyzed in the EEGs done immediately before and after tDCS on the first and last day of tDCS. One child remained seizure free for a year after tDCS. One child had reduced frequency of ICU admissions for status epilepticus for 2 weeks, likely due to reduced severity of seizures. In four children, an improvement in alertness and mood was reported for 2–4 weeks after tDCS. There was no benefit following tDCS in the other children. There were no unexpected or serious adverse effects in any child. Benefit was seen in two children, and the reasons for the lack of benefit in the other children need further study. It is likely that tDCS stimulus parameters will need to be tailored for different epilepsy syndromes and etiologies.

## 1. Introduction

Up to 30% of adults and children with epilepsy have drug-resistant epilepsy (DRE) [1,2]. For those who are not candidates for surgical treatments, therapeutic options are often limited (e.g., ketogenic diet and vagal nerve stimulation) and may not achieve seizure freedom [3,4,5,6,7]. Non-invasive brain stimulation (NIBS) techniques such as tDCS are being evaluated for the treatment of epilepsy [8,9], and cathodal tDCS has been reported to reduce seizure frequency in adults and children with epilepsy [10,11,12,13,14,15,16,17,18,19,20,21,22,23,24,25,26,27]. No serious adverse effects have been reported after therapeutic NIBS for epilepsy [9,28].

Most studies have reported some improvement in seizures after tDCS in adults and children with focal epilepsy [11,12,14,16,17,21,25,26,27]. Positive results (a decrease in seizure frequency) were also reported in those with Lennox–Gastaut syndrome (LGS) [13], Rasmussen’s encephalitis [16,21], epilepsia partialis continua [29], epileptic spasms [10], and Ohtahara syndrome [19]. Only rare studies have reported the failure of tDCS to improve seizure frequency [30,31]. Follow-up has been relatively short, with most successful trials reporting improved seizure frequency over 1 month [11,12,13,14,17,25,32] and some at 2–6 months [20,21,27]. It is not clear if the improvement of seizure frequency is longer lasting and associated with improved function or quality of life.

Successful trials used a wide range of stimulus parameters. One to fourteen daily sessions were used [12,13,14,17,30,32], stimulus currents were from 1 to 2 mA, and the stimulus duration was 20–40 min for each session [12,13,17,25,32]. A distinctive tDCS protocol was used by a few studies with a session consisting of two bouts of 9–20 min separated by 20 min [11,14,27,33]. Two studies used two or three cycles of tDCS (5–6 days of tDCS in each cycle repeated every 1–2 months) over 2–6 months [26,27]. Most studies have positioned the cathode over the region with the most intense EEG epileptiform activity [12,14,16,17,30,32]. This was found to be more effective than other areas of stimulation [10]. Two studies used customized montages of up to eight electrodes (multichannel tDCS) designed to simultaneously apply maximal cathodal stimulation over the seizure focus using modelling based on stereo EEG or surface EEG [25,27]. Anodal positions have varied in different studies and included the contralateral supra-orbital region, contralateral shoulder, and contralateral hemisphere [11,12,14,17,21]. The two trials reporting negative results used high-density tDCS (HD-tDCS) [30,31], although another study reported a benefit with HD-tDCS [29]. Thus, the relationship between the stimulus parameters and the magnitude and duration of the benefit is not clear [9].

There are fewer studies in children. In a randomized trial, Auvichayapat et al. [12] found a marginal but significant decrease in seizure frequency after a single tDCS session of 1 mA cathodal stimulation 20 min long in children with DRE. In children with LGS, tDCS (2 mA for 20 min) over five consecutive days was found to reduce seizure frequency up to 4 weeks [13]. Shelyakin et al. [18] reported a marked decrease in seizure frequency up to 6 months after tDCS (one to 15 sessions, 0.3–0.7 mA for 20–40 min) in children with cerebral palsy and epilepsy. Yang et al. [10] found tDCS to be more effective in children (than adults) with epileptic spasms. Yook et al. [15] reported a reduction in seizure frequency for 2 months in a child with cortical dysplasia after 10 sessions of tDCS. Two studies using multichannel tDCS showed an improvement of seizure frequency for up to 6 months in children and adults with refractory focal epilepsy [25,27].

Due to the variability of tDCS efficacy and parameters of stimulation in previous studies and the lack of clarity in the optimal parameters, especially in children, we undertook a pilot study to assess the tolerability and improvement in seizure burden and quality of life in children with DRE to guide in planning a controlled trial. Changes in the interictal epileptiform discharges were also evaluated by recording video EEG (vEEG) immediately before and after tDCS.

## 2. Material and Methods

An open label case study was performed on 12 children, aged 3 to 17 years, with DRE. The study was conducted according to the guidelines of the Declaration of Helsinki, approved by the Child and Adolescent Health Service Human Research Ethics Committee (Approval No. RGS0000002430) and registered with the Australian New Zealand Clinical Trials Registry (ACTRN12618001821280). The children were recruited from the neurology clinics of Perth Children’s Hospital and Princess Margaret Hospital for Children. Written informed consent was obtained from their parents or guardians (and assent from the children when appropriate). The inclusion criteria were children older than 2 years with DRE. DRE was defined as a failure of adequate trials of two or more tolerated and appropriately chosen and used antiepileptic drug schedules (whether as monotherapies or in combination) to achieve sustained seizure freedom [34]. The exclusion criteria were children less than 2 years, those with skin inflammation/allergies affecting the scalp, children in whom EEG could not be undertaken, and children with significant neuro-behavioral disorders.

### 2.1. tDCS

All patients received active tDCS treatment. Previous studies have shown benefits with many different stimulation protocols. We chose a simple protocol (for logistical reasons and its applicability in different clinical settings and varying resources) that was similar to that used by several previous successful trials [12,13,14,18]. The treatment consisted of three or four consecutive days of cathodal tDCS delivered using a NeuroConn DC stimulator MC (neuroConn Gmbh, Ilmenau, Germany). Although a 4-day course of tDCS was initially planned, it was not always possible due to the availability of children and EEG/tDCS facilities during any given week. Square surface electrodes (25 cm^2^) coated with EEG electrode gel were used for tDCS. The cathodes were applied over one or two EEG sites (international 10–20 system) with the most frequent epileptiform activity seen in ictal and interictal EEG recordings (Figure 1) based on assessments made by a child neurologist experienced in reading children’s EEGs. Anodes were positioned over the ipsilateral or contralateral supra-orbital region of the forehead depending on comfort and accessibility during EEG recording. Stimulation intensity and duration were 0.8 mA for younger children (<10 years) or 1 mA for older children (0.32–0.4 A/m^2^) for 20 min. However, two of the children aged 12 and 13 years received 0.8 mA of tDCS, instead of 1 mA of tDCS as planned. All children had video EEGs for 20 min before starting tDCS, during tDCS, and immediately after tDCS, on the first and last day of tDCS. No changes were made to antiseizure medications (ASMs) throughout the study. A tDCS questionnaire (see attachment 1) was completed by the child (or parent) before and after each session of tDCS.

### 2.2. Outcome Measures

Parents were asked to complete a Quality of Life in Childhood Epilepsy Questionnaire [35] (QOLCE, see attachment 2) before starting tDCS [35]. Parents were requested to keep a seizure diary (see attachment 3) 2 weeks prior to, during, and 2 weeks after tDCS. A neurological review was undertaken at 3 months and 6 months after tDCS, which included a structured interview (see attachment 4) of the parents and children to document the overall impression of changes in seizure frequency and severity, quality of life, behavior, motor and verbal performance, and sleep after tDCS.

### 2.3. EEG

Video EEG (vEEG) was recorded digitally, using the Compumedics equipment and Profusion software (Compumedics Limited, Victoria, Australia). Electrodes were applied in accordance with the International 10–20 system. Assessments of previous vEEG recordings (standard one-hour daytime recordings and prolonged 1- to 4-day vEEG studies in the epilepsy monitoring unit) by an epileptologist (LN) experienced in reading EEGs were used to determine the most active focus of the epileptiform activity (interictal and ictal). This was used to determine the application of the cathodal electrode over one or two locations. Two locations were chosen for children who had more than one focus of frequent epileptiform discharges on EEG.

vEEGs were recorded on the first and last day of tDCS (20 min before, during, and 20 min after tDCS) (Figure 1). The EEGs were reviewed for qualitative assessment (background, state, and epileptiform activity) and quantitative analysis of spikes (spike wave index, SWI) by neurologists (LN and SG) experienced in reading children’s EEGs. The SWI was calculated by observing spikes in the derivation with the most frequent spike activity in 15 s epochs of EEG. Epochs with and without spikes were counted. The SWI was equal to the number of epochs with spikes divided by the total number of epochs reviewed (SWI = number of epochs with spikes/number of epochs with spikes + number of epochs without spikes) and expressed as a percentage. SWI% was calculated separately for the awake/drowsy and sleep states, in the EEG before and after tDCS. There was some artefact in the EEG recorded during tDCS making the SWI assessment unreliable.

### 2.4. Statistical Analysis

The statistical analysis was performed only on SWI% measures in EEG before and after tDCS on the first day using SPSS software (SPSSv20 for Windows, SPSS Inc, Chicago, IL, USA). The Shapiro–Wilk test was applied to check for normality of the data. The statistical analysis was performed on SWI% before and after tDCS on the first day using a non-parametric test (Wilcoxon signed-rank test). The significance was set to <0.05. Statistical testing was not performed to compare the SWI% on the first and last day of tDCS due to the difference in the number of tDCS days between children.

## 3. Results

The clinical details of the 12 children in the study (six boys and six girls) are described in Table 1. All children had DRE. Their ages varied from 3 to 17 years, and most had seizure onset within the first 2.5 years of life. Most children (9/12) were developmentally delayed, and two were severely disabled (cases 4 and 9). The majority had focal or multifocal epilepsy with varying etiology including cortical dysplasia, post-infection, post-surgical scarring, and genetic cause. In two children, the diagnosis was Rassmussen’s encephalitis, and no cause had been found in two others. Most had frequent daily seizures, and seizure types included focal seizures, focal seizures with secondary generalization, and generalized seizures. Two children (cases 4 and 7) required frequent hospital and ICU admissions for convulsive status epilepticus (CSE). All children had been trialed on multiple ASMs (6–14), and most had tried one or more other interventions. These included steroids, folinic acid, pyridoxine, biotin, intravenous immunoglobulin (IVIG), ketogenic diet, and vagal nerve stimulation (VNS). Two children had undergone epilepsy surgery (cases 7 and 12), and another had surgery for craniosynostosis (case 11). The overall quality of life (question 9.1 on the QOLCE) was reported as good by the parent(s) of five children, fair for three, and poor for four.

### 3.1. Stimulus Parameters

The stimulus parameters and clinical outcomes (seizure frequency and clinic reviews) are detailed in Table 2. All children were on one or more ASMs at the time of tDCS (range one to five, mean 2.75). Cathodes were positioned over one or two of the EEG coordinates that showed the most frequent epileptiform discharges. Anodes were located over the ipsilateral or contralateral (and midline) supraorbital regions.

One child (case 7) had a seizure before the first day of tDCS and a seizure before and after the last day of tDCS (based on the vEEG recording). This was not unexpected as he had been having innumerable daily seizures at that time, and the seizure semiology was the same. None of the other children had any significant adverse effects during or after tDCS. Four children had mild irritation or itchiness during the stimulation, three reported a mild transient headache after the stimulation, and one child said he felt slightly wobbly on his feet for a few minutes after the stimulation.

### 3.2. Outcomes

Outcomes are detailed in Table 2. One child (case 12) remained seizure free for a year after tDCS, but seizures returned after that. One child (case 4) had frequent admissions to the ICU due to convulsive status epilepticus (CSE) prior to tDCS. Convulsive status epilepticus is defined as a generalized tonic–clonic convulsion lasting more than 30 min or repeated tonic–clonic convulsions over a 30 min period without the recovery of consciousness between each convulsion. She had fewer days of admission in the ICU for CSE for 2 weeks after tDCS, probably due to the reduced severity of CSE. As a result of this unexpected benefit, she received repeated monthly tDCS for 2 months (on a compassionate basis) after the first treatment, resulting in continuing benefits (reduced frequency of CSE) for 6 months. Based on her medical records, she spent 21 days in the ICU 6 months prior to tDCS and 8 days for the next 6 months after tDCS. Parents of four children (cases 1, 3, 7, and 12) reported a transient improvement in alertness and mood for 2–4 weeks. There was no benefit in the other children. At the clinic review at 3 and 6 months, parents reported improved QOL in two children (cases 4 and 12). There was no obvious difference in the outcomes between those that received 3 or 4 days of tDCS. None of the children (or their parents) reported any worsening of seizure frequency or severity after tDCS, nor did they report any other adverse effects.

### 3.3. EEG and Spike Wave Index

The spike wave index (expressed as a percentage, SWI%) was calculated in the EEG before and after tDCS, and the results are shown in Table 3. Most of the recordings were in the wake state, and further analysis was performed for SWI% values in the wake recording (Figure 1A,B). Figure 2 shows line graphs of the values at different times, and there was a clear trend for reduction in SWI% in most children from before to after tDCS on the first and last day. In a few children (cases 2, 4, 5, 11, and 12), there was reduction in SWI% from the post-tDCS value on the first day to the pre-tDCS value on the third or fourth day. In the three children with SWI% values of 100% before and after tDCS, a reduction in the spike density was observed after tDCS (on visual analysis) but was not reflected in the SWI% values due to the high spike burden in the EEG (see Figure 1C,D).

Non-parametric statistical analysis of the SWI data before and after tDCS on the first day showed a significant reduction in the SWI% after tDCS (Wilcoxon signed-rank test, Z = −2.599, *p* = 0.009).

An increase in the background β activity was noted in the EEGs after tDCS in most children, and in one child, there was less asymmetry after tDCS (Table 3).

## 4. Discussion

Many studies have explored the use of NIBS for the treatment of seizures in patients with DRE [8,9,28]. Cathodal tDCS and low-frequency repetitive Transcranial magnetic stimulation (LF-rTMS) have been used to reduce cortical excitability underlying the stimulating electrodes [8,28,36]. The neural mechanisms by which TMS and tDCS affect cortical excitability are different; while TMS causes depolarization of neurons in the underlying cortex and can elicit action potentials, tDCS causes subthreshold depolarization or hyperpolarization of neurons [8,28]. The long-term effects of rTMS and tDCS are believed to be mediated by neuroplastic mechanisms, by modulating long-term potentiation (LTP) and long-term depression (LTD) like phenomena [8,28].

Most studies using cathodal tDCS have shown benefits in adults and children with focal epilepsy. However, tDCS treatment in our study showed benefits in only two children, even though our treatment parameters were similar to those in many previous studies.

There are many limitations of the study. This was a pilot study to help plan a controlled trial and therefore did not include a placebo/sham arm. Many previous investigators have performed pilot studies to evaluate the tolerability, electrophysiological markers, and effectiveness of tDCS to optimize the design of subsequent controlled trials [8,26,27]. The sample size was small but similar to previous open label trials and based on sample size calculations for pilot studies [8,25,26,27,37]. We included children with varying ages and durations of seizures to assess the safety and tolerability in a wide variety of epilepsies. Many of the children had innumerable seizures every day, making accurate counts of seizures difficult. The clinical profile of the children varied considerably, as did the severity and frequency of seizures. In some children, the frequency, severity, and types of seizures varied considerably from week to week and month to month, making it difficult to establish a clear baseline over short periods. We tried to overcome this by reviewing them in the clinic at 3 and 6 months to confirm changes, if any, were reliably present over a longer duration.

In our study, there was seizure freedom in one child for a year after tDCS (case 12), and one child (case 4) had fewer ICU admissions for CSE for 2 weeks after tDCS, probably due to the reduced severity of seizures. The only child to have a long-term benefit in our study also had the lowest frequency of seizures at baseline. There was no meaningful benefit for the other children, although parents reported transient improvements in alertness and mood in three of them. This result most closely matches that of Auvichayapat et al. [12], who found a marginal but significant decrease in seizure frequency after a single tDCS session. The increase in the number of tDCS sessions in our study did not appear to improve the outcome. We observed a significant reduction in the EEG epileptiform discharges after tDCS that was also reported by them. Other studies [15,18,19] have reported better outcomes with treatment protocols similar to ours. For example, Auvichayapat et al. [13] found a 90% decrease in seizure frequency for over a month in children with Lennox–Gastaut syndrome after five sessions of 2 mA tDCS, Shelyakin et al. [18] found a decrease in seizure frequency for up to 6 months following one to 15 sessions of 0.3–0.7 mA cathodal tDCS, and Fregni et al. [32] showed a 44% reduction in seizure frequency for 1 month following one session of 1 mA cathodal tDCS. There are several reasons why we may have been unable to show similar benefit. Many of the children in our study had multiple seizure types that appeared to be less responsive to tDCS [10]. tDCS may be less effective in children with epileptic encephalopathies [31]. The high seizure burden in some of the children (innumerable seizures per day) may have made a modest reduction unverifiable. Most children were on multiple ASMs, which may have interacted with tDCS. Other successful studies have used multichannel tDCS (based on modelling), more tDCS sessions, or repeated cycles of tDCS, which may be more effective in controlling seizures [8,10,25,26,27].

The pilot study also suggested ways in which we may be able to overcome some of these limitations in future trials. We plan to use a controlled study design to overcome the limitations including the variability in seizure frequency over days, weeks, and months. We plan to use a crossover design so that each patient acts as his or her control. We plan to use other scoring tools to look for a reduction in epilepsy burden apart from seizure frequency and guided by the seizure patterns (for example, the frequency of the use of rescue medications for prolonged seizures, the number of school days missed due to seizures, the number of injuries sustained after convulsive seizures, and the number of seizure-related admissions to hospital). We plan to recruit children with DRE who have a homogenous clinical profile in a single study (for example, children with normal development and focal epilepsy or children with a specific type of developmental and epileptic encephalopathy). Protocols need to be more carefully planned (based on the availability of resources and patient convenience) so that there is better adherence to stimulation parameters (for example, intensity, duration, and location) and outcome measures. Other features of EEG apart from epileptiform discharges (for example, background changes and changes in functional connectivity) may be useful in understanding tDCS effects and outcomes. It would be useful to trial newer tDCS protocols that may have a longer effect such as a single session consisting of two bouts of 9–20 min separated by 20 min [11,14,27,33]).

The effects of tDCS on epilepsy surrogate markers have been commonly obtained from the frequency of EEG epileptiform discharges. Our study showed an immediate reduction in EEG epileptiform discharges after tDCS. Some studies have reported reductions in EEG epileptiform discharges for up to 4 weeks after a single or multiple tDCS sessions [12,13,32]. Other studies have found no changes in EEG epileptiform discharges after tDCS [17,21,26]. A reduction in epileptiform activity has not been uniformly related to a reduction in seizure frequency, as was also seen in our study. It is possible that the improvements in EEG epileptiform discharges in our study were transient and therefore not beneficial in changing seizure outcomes. Changes in background EEG after tDCS were observed in this study and have been reported in previous studies [18]. We found an increase in background β activity in 10 of the 12 children. The amount of beta activity is thought to reflect increased vigilance and alertness [38]. The effects of background EEG changes following tDCS have not been explored. Other studies have evaluated functional connectivity in different frequency bands (alpha, beta, and theta bands) and related reductions in connectivity to seizure reduction after tDCS [26,27].

The few studies that have assessed QOL did not find any improvement after tDCS despite the reduction in seizure frequency [11,12]. In our study, four children were observed to be more alert and happier after the tDCS for a short period, but an improvement in QOL was reported in only two children (cases 4 and 12). None of the children in our study reported any serious adverse symptoms after tDCS, and it appears to be safe and well tolerated [9,39]. Although a seizure was observed after tDCS in one child in our study, this likely reflected the seizure burden at that time. Seizures have been reported by others during or after tDCS in children with frequent seizures [11,21].

There are some limitations of the therapeutic uses of tDCS. Although there are demonstrable changes in cortical excitability for up to an hour after a single session of tDCS, the effects of repeated stimulation over days and weeks are less well understood [40,41]. The effects of tDCS (i.e., excitatory or inhibitory) may be affected by the polarity and intensity of stimulation [42]. Although adverse effects are usually mild, seizures have been rarely reported to occur during tDCS, and there may be worsening of seizure frequency in a few children after tDCS [11,21,25]. A potential limitation of tDCS is its limited spatial accuracy. As the current passes through the brain from the cathode to the anode and modulates neural activity simultaneously underneath the cathode and the anode, it can be difficult to relate the effects of tDCS to a specific brain region. Thus, tDCS not only affects the brain regions directly under the electrodes but may also modulate functional connectivity between remote but functionally associated brain areas or influence the within-network connectivity [43].

## 5. Conclusions

This open label pilot study suggests that the role of tDCS in DRE in childhood, particularly in those with a high seizure burden, remains unclear. tDCS is well tolerated and appeared to make some children (30%) happier and more alert for a brief period. Impacts on seizure frequency or severity were seen in two (17%); one child became seizure free for a year, and another had fewer ICU admissions for 2 weeks. There was a significant reduction in SWI between the pre-tDCS and post-tDCS EEGs on the first and last day of tDCS. This reduction did not lead to the reduction in seizure frequency, as has been reported before [12]. Our study adds to the literature that suggests tDCS may have a promising role in DRE, but further study is needed to find the optimal stimulus paradigm for the different epilepsy etiologies and electroclinical syndromes.

## Figures and Tables

**Figure 1 brainsci-13-00760-f001:**
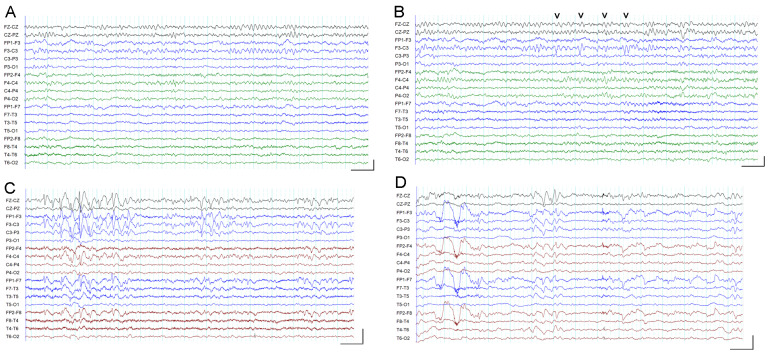
EEG recordings. All EEG epochs are in a bipolar double banana montage. Calibration bars in each subfigure: horizontal, 1 s; vertical, 0.2 mV. (**A**,**B**) The 15 s epochs from wake EEG recording from case 12 before tDCS showing no epileptiform activity in (**A**) and epileptiform discharges in the F3-C3 derivations with phase reversal at C3 (see arrowheads) in (**B**). (**C**,**D**) The 15 s epochs from wake EEG recordings from case 8 before (**C**) and after (**D**) tDCS. Epileptiform discharges are seen in both epochs; however, there is a reduction in the amplitude and frequency of epileptiform discharges in (**D**). The symbols “v v v v” are the arrowheads referred to in description of Figure 1B. The horizontal and vertical bars at the bottom right corner of each subfigure are calibration bars mentioned in the caption.

**Figure 2 brainsci-13-00760-f002:**
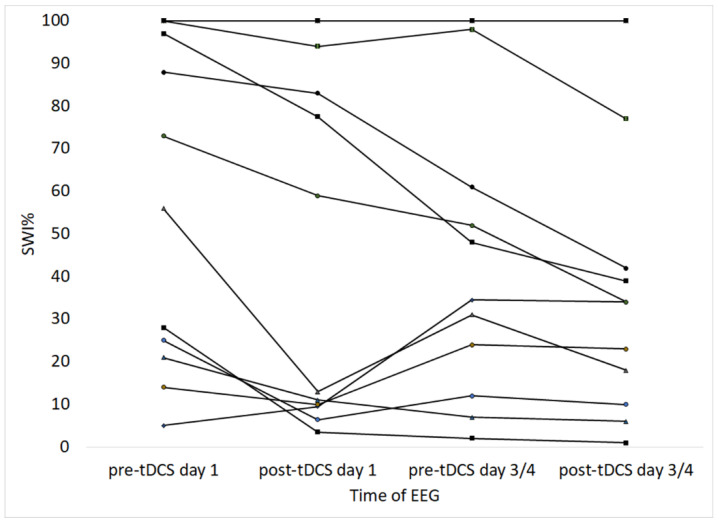
Line graphs showing the spike wave index percent (SWI%) before and after the first and last (third or fourth) day of tDCS in individual cases. Abbreviations: SWI: spike wave index; tDCS: transcranial direct current stimulation.

**Table 1 brainsci-13-00760-t001:** Clinical details of children participating in the trial.

Case No. and Gender	Age at tDCS (Years)	Age of Onset of Seizures	Seizure Types	Interictal Epileptiform Discharges	Epilepsy Classification	Etiology of Epilepsy	Development	No. of ASMs Tried	Other Treatments	Seizure Frequency
1F	17	3 m	FS, FS with SG	Focal RH	FE	Cortical dysplasia	Delayed	13	Steroids, folinic acid, pyridoxine	Variable, often >20/day
2M	17	2 y 5 m	GTC, AtS, FS	Focal RH > LH	FE	Rassmussen’s forme fruste	Delayed	9	IVIG, ketogenic diet	Variable, upto 80/day
3F	17	12 y	GTC, MS, AS, FS, NCSE	Generalized, focal occ RH + LH	GGE + IPOE	Not identified	Regression	11	VNS, ketogenic diet	Variable, two to three GTCS/day, innumerable MS/day
4F	4	3 m	GTC, AAS, MS, FS, CSE	Generalized, multifocal	FE + GE	Dravet syndrome	Delayed	11	Pyridoxine, prednisolone, ketogenic diet, IVIG, biotin	Innumerable per day, frequent status, in and out of hospital
5F	3	6 y	FS, FS with SG	Focal RH	FE	Rassmussen’s form fruste	WNL	6	Regular IVIG every 4 weeks, alternating with pulsed dexamethazone for 3 days every 4 weeks	25–30/day
6F	3	2 y 2 m	FS, FS with SG	Focal RH, ESES RH	FE	Cortical dysplasia	WNL	7	None	>50/day
7M	3	1 y 2 m	FS, TS, FS with SG	Focal LH	FE	Cortical dyspasia	Delayed	16	Ketogenic diet, IVIG, IV methyl prednisolone, epilepsy surgery	Innumerable seizures per day, frequent status, in and out of hospital
8F	10	2 y 2 m	FS, TS, AtS, AAS	Multifocal LH > RH	FE	HSV encephalitis	Delayed	14	VNS, steroids, IVIG	10 seizures per month, up to two to three per day
9M	7	6 m	FS, TS, AAS, GTC	Multifocal RH + LH	FE + GE	Not identified	Delayed	7	None	30 seizures per month
10M	12	8 m	FS, FS with SG, HS	Focal LH	FE	Cortical dysplasia	Delayed	11	None	From one per week to one to two per day
11M	16	12 y 6 m	FS, AAS	Multifocal RH + LH	FE	Focal scarring following cranial surgery	WNL	9	None	One to two per day
12M	13	10 m	FS, FS with SG	Focal LH	FE	Cortical dysplasia	Delayed	13	Epilepsy surgery	One per week

Abbreviations: Seizure types: AAS: atypical absence seizure; AS: absence seizure; AtS: atonic seizure; CSE: convulsive status epilepticus; FS: focal seizure; FS with SG: focal seizure with secondary generalization; GTCS: generalized tonic–clonic seizure; HS: hyperkinetic seizure; MS: myoclonic seizure; NCSE, non-convulsive status epilepticus; TS: tonic seizure. Other abbreviations: ESES: electrical status epilepticus of sleep; IV: intravenous; IVIG: intravenous immunoglobulin; LH: left hemisphere; RH: right hemisphere; VNS: vagal nerve stimulation; WNL: within normal limits.

**Table 2 brainsci-13-00760-t002:** Clinical features at tDCS, stimulus parameters, and outcomes in all children.

Case No. and Gender	Age at tDCS (Years)	Antiseizure Medications at tDCS	Duration of Epilepsy at tDCS (Years)	Days of tDCS	Stimulation Amplitude (mA)	Cathode Location	Anode Location	No. of Seizures in 2 Weeks Pre-tDCS	No. of Seizures in 2 Weeks Post-tDCS	Other Improvements After tDCS at 2 Weeks	Adverse Effects of tDCS
1F	17	LCM, OXC, ZON	17	4	1	F4	LSO	4	4	More alert, happier, less moody	Headache, irritation
2M	17	LCM, TPM	7	4	1	P4, T4	RSO, midline SO	Innumerable (up to 80/day)	Innumerable	None	Irritation
3F	17	CLON, LTG, LEV, TPM, VPA	7	3	1	C1, C2	LSO, RSO	~40	~40	Happier and spoke more clearly	Headache
4F	4	TPM, PB, RUF, LEV	3	3	0.8	C1, C2	LSO, RSO	Innumerable	Innumerable	Reduced ICU admissions for status for several months, more alert	None
5F	3	LTG, PER, immunotherapy	7	4	0.8	C4	RSO	~20	~20	None	Itchiness
6F	3	SUL, LEV	1	4	0.8	C4, T4	RSO, midline SO	>50	>50	None	Irritation
7M	3	LTG, RTG, PTY, NZP	2	3	0.8	F3	LSO	Innumerable	Innumerable	None	Seizure
8F	10	ZON, CLOB, LCM	8	4	1	F3, F4	LSO, RSO	7	6	None	None
9M	7	VPA	7	3	0.8	F5, F6	LSO, RSO	13	14	More alert, sleeps better, walking better	None
10M	12	LEV, LTG	11	4	0.8	F3	LSO	14	13	None	Unsteady gait
11M	16	SUL, LTG	4	3	1	F3, F4	LSO, RSO	~20	~20	None	None
12M	13	PB, LCM, CBZ	13	4	0.8	Fp1, F3	LSO, midline SO	1	0	More alert, playing sport	Headache

Abbreviations: ASMs: antiseizure medications; CBZ: carbamazepine; CLB: clobazam; CLON: clonazepam; LCM: lacosamide; LTG: lamotrigine; LEV: levatiracetam; NZP: nitrazepam; OXC: oxcarbazepine; PER: perampanel; PB: phenobarbitone; PTY: phenytoin; RTG: retigabine; RUF: rufinamide; SUL: sulthiame; TPM: topiramate; VPA: valproic acid; ZON: zonisamide. Other abbreviations: LSO: left supraorbital; SO: supra-orbital; RSO: right supra-orbital. EEG coordinates based on the 10–20 system: C1, C2, C4, Fp1, F3, F4, F5 (between F3 and F7), F6 (between F4 and F8), P4, T4.

**Table 3 brainsci-13-00760-t003:** Spike wave index percent (SWI%) values and other observations in EEG recordings taken before and after tDCS on the first and last day of tDCS.

Case No.	First Day of tDCS					Last Day of tDCS				
and	Before tDCS		After tDCS			Before tDCS		After tDCS		
Gender	Awake	Sleep	Awake	Sleep		Awake	Sleep	Awake	Sleep	
	SWI%	SWI%	SWI%	SWI%	Other Observations	SWI%	SWI%	SWI%	SWI%	Other Observations
1F	25		6.4		↑β activity post-tDCS	12		10		↑β activity post-tDCS
2M	88		83		↑β activity post-tDCS	61		42		
3F	100	100				100		100		↓spike density post-tDCS
4F	97		77.5		↑β activity post-tDCS	48		39		
5F	28		3.5			2		1		↑β activity post-tDCS
6F	100	100	94		↑β activity, ↓spike density post-tDCS	98		77		
7M	5	12.5	9.5	10	Seizure before tDCS	34.5		34		Seizure before and after tDCS
8F	100		100		↑β activity, ↓spike density post-tDCS	100		100		↑β activity, ↓spike density post-tDCS
9M	56		13		↑β activity, ↓spike density post-tDCS	31	67	18		↑β activity, ↓spike density post-tDCS
10M	14		10		↓spike density post-tDCS	24		23		↑β activity post-tDCS
11M	21		11		↑β activity post-tDCS	7		6		
12M	73	80	59		↑β activity, less asymmetry post-tDCS	52	57	34	38	↑β activity, less asymmetry post-tDCS

Abbreviations: SWI: spike wave index; tDCS: transcranial direct current stimulation.

## Data Availability

No other data are available due to patient privacy.

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
