# Peer review of "Tolerability and Effectiveness of Cathodal Transcranial Direct Current Stimulation in Children with Refractory Epilepsy: A Case Series"

_brainsci, 2023, doi:10.3390/brainsci13050760_

Round 1
Reviewer 1 Report (Previous Reviewer 1)
No further comments. Thank you.
Reviewer 2 Report (Previous Reviewer 2)
After revision the manuscript has been significantly improved
This manuscript is a resubmission of an earlier submission. The following is a list of the peer review reports and author responses from that submission.
Round 1
Reviewer 1 Report
This is very interesting and well written manuscript about an important topic.
However, I have some minor suggestions and questions.
1. Please justify a bit more the stimulation parameters for this study.
2. Justify the absence of a SHAM group.
3. There is a wide age range (3-17 years). Any influence of age on the results? For example, hormonal changes during puberty. Please comment.
4. tDCS has limitations. Include this in the discussion.
Reviewer 2 Report
At the manuscript "Tolerability and effectiveness of cathodal transcranial direct current stimulation in children with refractory epilepsy: a case series” by Drs. Soumya Ghosh and Lakshmi Nagarajan authors reported results of the study aimed to investigate the tolerability and effectiveness of cathodal transcranial direct current stimulation (tDCS) in drug resistant epilepsy (DRE) in children. Some patients showed improvement, in rare cases worsening, no side effects were found.
These results fit well into the overall picture of the practice of using tDCS.
The authors conducted an interesting study and received significant data, I have no objection to the essence of the study. It is necessary to agree with the conclusions of the authors, who believe that the role of tDCS in DRE in childhood, especially in individuals with a high frequency of seizures, remains unclear. At the same time, tDCS is well tolerated and in some cases improves the quality of life.
The pilot study does not require a detailed discussion, but still I have a question:
The authors rightly mention the parallels between transcranial magnetic stimulation and tDCS. Is there any data on comparing the biological mechanisms of action of these methods? I think it would be right to add at least a couple of phrases about this.
Minor criticism:
LINE 35: evaluated for treatment of epilepsy [8, 9]. and cathodal tDCS has been reported to reduce
I think there is a typo here, do not need a dot before "and"
The authors explain the abbreviation “LGS” (Lennox-Gastaut Syndrome (LGS)) twice, once is enough
The presentation of a subject is systematic and comprehensive and analysis is proper.